# Social Media Use and Adolescents’ Sleep: A Longitudinal Study on the Protective Role of Parental Rules Regarding Internet Use before Sleep

**DOI:** 10.3390/ijerph18031346

**Published:** 2021-02-02

**Authors:** Regina J.J.M. van den Eijnden, Suzanne M. Geurts, Tom F.M. ter Bogt, Vincent G. van der Rijst, Ina M. Koning

**Affiliations:** Interdisciplinary Social Science, Utrecht University, Padualaan 14, 3584 CH Utrecht, The Netherlands; s.m.geurts@uu.nl (S.M.G.); T.F.M.terBogt@uu.nl (T.F.M.t.B.); v.g.vanderrijst@uu.nl (V.G.v.d.R.); i.koning@uu.nl (I.M.K.)

**Keywords:** social media use, social media addiction, sleep, parental rules, parenting, adolescents

## Abstract

The popularity of social media use among adolescents has raised concerns about the potentially harmful effects of social media use on adolescents’ sleep. Since longitudinal research considering this relationship is scarce, the present two-wave longitudinal study of 2021 secondary school students (M_age_ = 13.86, SD = 1.25) examined whether frequency of social media use and problematic social media use predicted adolescents’ bedtime and quality of sleep. Moreover, the protective role of parental rules regarding Internet and smartphone use one hour before sleep was examined. The findings indicated that strict parental rules about Internet and smartphone use before sleep might prevent negative consequences of social media use on bedtime and sleep quality, but only among less engaged social media users. Once adolescents are highly engaged social media users, strict parental rules do not seem to prevent negative media influences on sleep. This implies that limiting Internet accessibility before bedtime can help prevent adolescents’ sleep problems, but that regulation is less effective for adolescents who are already highly involved in social media use.

## 1. Introduction

Using social media—such as Facebook, Instagram, Snapchat, and WhatsApp—has become one of the most popular leisure-time activities among adolescents. In 2017, 98% of Dutch adolescents aged 12–15 years owned a smartphone, and about 31% agreed they are using social media on their smartphone throughout the whole day [1]. The increasing amount of time adolescents spend using social media has raised concerns about its potential negative influence on adolescents’ health and well-being, including sleep [2,3]. Sleep is vital for adolescents’ learning capacity, memory processes, emotional regulation, and related behaviors [4,5]. Insufficient or disturbed sleep is related to attention problems [6], poor school performance [7], daytime tiredness [8], depression [9], and obesity [10]. Unfortunately, sleep problems among adolescents are rather common, with 10% of Dutch adolescents aged 14–17 years sleeping less than 6.5 h, 16.3% experiencing difficulties in falling asleep, 24.1% experiencing sleep interruptions, and 44.3% reporting feeling not rested upon waking up [11]. Increasing the knowledge about predictors of sleep problems among adolescents is, therefore, warranted.

Numerous studies have linked sleep problems with screen-based activities in general (for review, see [12]). However, longitudinal research on sleep and social media use among adolescents is limited. Since social media channels have become a fundamental part of adolescents’ daily lives, it is vital to investigate whether their use affects adolescents’ sleep. Therefore, the current longitudinal study examined the effects of social media use on adolescents’ actual bedtime and their perceived quality of sleep. Furthermore, we tested whether parental rules regarding Internet use in the hour before going to sleep and parental rules about taking the smartphone to the bedroom when going to sleep moderate these effects. Moreover, several scholars have emphasized that frequency of social media use and problematic social media use should be regarded as related but different dimensions [13,14,15]. Problematic social media use can be defined as a loss of control over one’s social media use, accompanied by significant impairments in daily-life functioning [13,14]. International research data show that 4% to 18% of 15 year olds from 42 countries and regions reported high levels of problematic social media use in 2017–2018 [16]. It is important to distinguish between frequency of use and problematic use because frequent users may be well able to regulate their social media use, whereas problematic users may not show intensive patterns of use, for instance, due to a mismatch between their desired and actual online social network [13,14].

### 1.1. Frequency of Social Media Use and Sleep

Previous cross-sectional studies have consistently shown that the frequency of social media use is associated with several sleep problems among adolescents. Adolescents who frequently use social media during the day report less total sleep time [17] and poorer sleep quality [18]. The frequency of social media use, particularly in the evenings, is related to a delay in bedtime [19], more difficulties falling asleep [20,21], a shorter sleep duration [20,22], and a poorer sleep quality [18,23]. Several mechanisms may play a role in these relationships. First, phone, tablet, or laptop screens emit blue light, which may suppress the production of the sleep-promoting hormone melatonin (e.g., [24,25]). Second, online activities (e.g., chatting) just before bedtime may cause emotional, mental, and/or physiological arousal [26], which may, in turn, lead to more difficulty falling asleep and poorer sleep quality [27]. Third, because social media use does not include a defined endpoint, adolescents may find it difficult to stop. Social media use in the evenings may postpone going to sleep and instigate irregular bedtimes, which may affect sleep duration and sleep quality [3]. Finally, social media might cause sleep interruptions when adolescents keep their smartphones in their bedroom at night, as social media may generate incoming messages accompanied by visual and sound alerts [18]. Yet, because of the cross-sectional nature of the mentioned studies, it is impossible to establish the direction of these relationships or their underlying mechanisms.

Van der Schuur, Baumgartner, and Sumter [28] conducted cross-sectional and longitudinal analyses regarding the link between the frequency of social media use and sleep-related variables, i.e., daytime tiredness and sleep latency. Their cross-sectional analysis showed that daytime tiredness was more prevalent in those with higher scores on social media use, in line with the previously mentioned findings. Additionally, more social media use was associated with longer sleep latency, i.e., the time between going to bed and falling asleep. However, longitudinal evidence for prospective relations between the frequency of social media use and daytime tiredness, as well as sleep latency, was not found [28]. As far as we know, this is the only available longitudinal study on the link between the frequency of social media use and adolescents’ sleep problems. Hence, longitudinal research on the effect of the frequency of social media use on other relevant sleep variables, such as time of going to sleep and perceived quality of sleep, is completely missing. The current longitudinal study addressed this gap. Based on the previous cross-sectional findings, we expected that more frequent use of social media would predict a later bedtime and a poorer sleep quality among adolescents.

### 1.2. Problematic Social Media Use and Sleep

Research on the relationship between problematic social media use (PSMU) and sleep outcomes is also rather scarce. A cross-sectional study among Australian adolescents aged 12–18 years found that problematic social networking was related to more sleep disturbances and poorer sleep quality [29]. Similarly, two cross-sectional studies of students with a mean age of 20 years demonstrated that Facebook dependence and social media addiction were associated with a poorer sleep quality [30,31]. Other cross-sectional research has addressed the link between more general forms of Internet addiction and sleep problems, indicating that Internet addiction among adolescents is related to longer sleep latency, increased use of sleep medication, lower sleep efficiency, more sleep interruptions [32], less total sleep time [33,34], poorer sleep quality [32,35], and more daytime tiredness [32,36]. Furthermore, a longitudinal study among children and adolescents aged 8–16 found that Internet addiction predicted a disturbed sleep–wake rhythm [37]. Based on these findings, we hypothesized that problematic social media use among adolescents would predict a later bedtime and poorer sleep quality.

### 1.3. The Role of Parental Rules Regarding Internet and Smartphone Use before Going to Sleep

It has been argued that the timing rather than the duration of social media use may affect adolescents’ sleep [15], as social media use at bedtime is a strong predictor of poorer sleep outcomes [38]. Thus, social media use during the evening hours and before going to sleep may be particularly associated with sleep problems. Therefore, it can be expected that the rules that parents set regarding Internet and smartphone use before sleep will buffer the assumed negative effect of social media use on sleep. For instance, it can be anticipated that adolescents who are allowed to bring their smartphone or tablet to their bedroom when they go to sleep will experience more sleep problems than adolescents who have to leave their smartphone or tablet behind.

The literature on the effects of parental rules about the Internet, social media, and smartphone use before bedtime has been limited. Moreover, studies on the effects of general Internet-specific rules on the intensity of adolescents’ Internet use and problematic use are rather inconsistent. On the one hand, several cross-sectional studies have found negative relationships of Internet-specific rules with time spent online [39,40] and problematic Internet use [41,42,43], suggesting that setting strict rules is effective in shaping adolescents’ online behavior. On the other hand, some cross-sectional studies did not reveal a relationship between Internet-specific rules and problematic Internet use [40] or time spent online [44], and one study even found a positive relationship, suggesting that strict parental rules may promote the development of problematic Internet use [45]. Concerning social media use, a longitudinal study showed that Internet-specific rules predicted fewer symptoms of problematic use among girls but not among boys [46].

As far as we know, two cross-sectional studies addressed parental rules regarding screen use in the evening hours. The study of Buxton et al. [47] among youth aged 6–17 years concluded that children whose parents enforce strict rules regarding the amount of time their child is allowed to use a smartphone, tablet, or computer in the evening slept longer compared to children whose parents impose such rules only sometimes or never. In addition, a cross-sectional study among Belgium adolescents aged 13–20 years showed that the presence of rules regarding media use in combination with the presence of rules regarding bedtime moderated the positive relationship between some types of media use (computer use, mp3-player use, and television viewing) in the evening and actual bedtimes [48]. Based on these studies, we expected that rules about Internet and smartphone use before bedtime would moderate the relationship between social media use and adolescents’ sleep. More specifically, we hypothesized that the assumed effect of social media use on sleep would be weaker when parents set strict rules about Internet and smartphone use before going to sleep.

### 1.4. Present Study

This two-wave longitudinal study aimed to contribute to our understanding of the effects of social media use on adolescents’ sleep and extend our knowledge of the preventive role that parents can play to protect adolescents from sleep-disturbing social media use. First, we examined the effects of both the frequency of social media use and problematic social media use on adolescents’ bedtime and adolescents’ perceived quality of sleep. Next, we tested whether parental rules about using the Internet in the hour before going to sleep and taking the smartphone to the bedroom moderated these effects. These relationships were tested in two separate models, one with the intensity of social media use and one with problematic social media use as the main predictors, and a combined model including both predictors simultaneously. Based on the previous empirical findings, we hypothesized that (1) both the frequency of social media use and problematic social media use will predict a delay in bedtime and poorer perceived quality of sleep and that (2) these effects will be mitigated when parents set strict rules regarding Internet and smartphone use before bedtime.

## 2. Methods

### 2.1. Sample

The data for this study were derived from the Digital Youth Project, a longitudinal research project on adolescents’ online behavior. Specifically, we used data from two measurement waves conducted in February 2017 and 2018 (T3 and T4 in the original project, in this study, referred to as T1 and T2), as some variables of interest were not included in the questionnaire in the first two waves of the original study. Adolescents from six different secondary schools completed a computer-based questionnaire during school hours. Research assistants were present to supervise the data collection, answer student questions, and ensure maximum privacy.

Of the 2708 participants included at T1, 678 dropped out of the study due to withdrawal of one of the six schools. Of the remaining 2030 participants, 1422 (70%) also participated at T2. Further dropout of participants was mainly due to dropout of complete classes (particularly last classes of lower-level secondary education) because schools did not provide permission for these specific classes, or, to a lesser extent, absence of individual students during the measurement days. Nine adolescents were excluded because of unreliable data, resulting in a final sample of 2021 participants eligible for analysis. More than half of the final sample was male (54.6%), and the age ranged from 11 to 17 years (*M_age_* = 13.86, *SD* = 1.25) at T1. Students were in lower-level vocational education (42.2%), moderate level secondary education (11.8%), or high school/pre-university education (46.1%). Most students (95.4%) had a Dutch background.

Participants who dropped out of the study were more likely to be older (*t*(2696) = −15.98, *p* = < 0.001), to be male (χ^2^ (1) = 6.36, *p* = 0.01), to follow a lower level of education (χ^2^ (1) = 340.88, *p* < 0.001), and to have parents who do not impose strict rules regarding using the Internet in the hour before going to sleep (χ^2^ (1) = 15.69, *p* < 0.001) or taking the smartphone to the bedroom when going to sleep (χ^2^ (1) = 37.90, *p* < 0.001) compared to participants who participated in both waves. Additionally, participants who dropped out were more likely to report more frequent social media use (*t*(2704) = 5.40, *p* < 0.001), more problematic social media use (*t*(2630) = −2.63, *p* = 0.008), later times of going to sleep (*t*(2500) = −9.69, *p* < 0.001), and poorer quality of sleep (*t*(2598) = 2.09, *p* = 0.04). These differences may to some extent be explained by the fact that several whole school classes from the last year of lower-level secondary education dropped out.

### 2.2. Measures

The frequency of social media use was measured using six items. A definition of social media was given preceding the questions [13]. Four items measured the use of social network sites (e.g., “How many times a week do you post a message, picture, or video on your social network sites?”), and two items measured instant messaging through smartphones (e.g., “How many times a day do you send a message, picture, or video with your smartphone?”) on a 7-point response scale ranging from 1 (less than once a day/week) to 7 (more than 40 times a day/week). A sum score was calculated, where a higher score reflects a more frequent use of social media. Cronbach’s α values were 0.84 (T1) and 0.82 (T2).

Problematic social media use was measured using the Social Media Disorder (SMD) scale containing 9 items [49] corresponding to the nine diagnostic criteria for Internet Gaming Disorder (IGD) according to the Appendix of the DSM-5. These criteria entail preoccupation, tolerance, withdrawal, persistence, escape, conflict, problems on important life domains, displacement, and deception. Because both IGD and SMD are not officially recognized as mental disorders by the DSM-5, and because there is an ongoing scholarly discussion on how to conceptualize behavioural addictions [50], we prefer to use the term problematic use over disordered use. Adolescents were asked the following question: “During the past year, have you (…)”, followed by, for example, “regularly had no interest in hobbies or other activities because you would rather use social media?”, which refers to the criterion “displacement”. Respondents replied on a dichotomous scale (1 = no and 2 = yes). The nine items were summed, with a higher score indicating more social media disorder symptoms. Given the items’ dichotomous nature, internal consistency was calculated using the ordinal alpha based on the tetrachoric correlation matrix [51]. Ordinal alpha values were 0.90 (T1) and 0.89 (T2).

Perceived quality of sleep was measured using the 5-item Groningen Sleep Quality Scale (GSKS; [52]) and one item from the Depressive Mood List: “I have trouble going or staying asleep” [53]. Response categories ranged from 1 (never) to 5 (always). Most items were recoded so that a higher mean score reflected the higher quality of sleep. Cronbach’s α values were 0.83 (T1 and T2).

Bedtime was measured with one item, “On schooldays, at what time do you go to sleep?” Response categories ranged from 1 (before 20.00) to 10 (after 00.00), with higher scores indicating later bedtime.

Two parental rules regarding Internet and smartphone use before bedtime were included as moderators. Adolescents indicated the extent to which they are allowed to (1) use the Internet or play games in the hour before they go to sleep and (2) bring their smartphone/tablet to their bedroom at bedtime on regular school days. Response categories ranged from 1 (never) to 5 (very often). These variables were recoded into dichotomous variables, with 0 = no strict rules (“sometimes” to “very often”) and 1 = strict rules (“never” or “rarely”), since the parental rules variables were not normally distributed.

### 2.3. Data Analyses

Descriptive statistics and correlations between all study variables were computed using IBM SPSS Statistics 24. Next, structural equation modeling was conducted using Mplus 8.1 [54] to examine the effects of social media use on adolescents’ sleep. The effects of frequency of social media use and problematic social media use at T1 were tested in two separate multivariate models, which included each predictor separately, and a comprehensive multivariate model that included both predictors simultaneously. In these models, we also controlled for gender, age, educational level, and the outcome variables at T1. To test the moderating role of parental rules about using the Internet in the hour before going to sleep and of parental rules about taking the smartphone to the bedroom when going to sleep, interaction terms between the predictor variables and these two parental rules were included in four separate models. The predictive value of these interaction terms was tested while controlling for the main effects of these moderators. Maximum Likelihood estimation was used to handle missing data. Bootstrapping (500) was applied because the predictor “problematic social media use” was not normally distributed. The Comparative Fit Index (CFI) and the Root Mean Square Error of Approximation (RMSEA) were used to evaluate model fit. According to standard criteria, CFI values equal to 0.90 or higher and RMSEA values below 0.08 indicate an acceptable model fit [55].

## 3. Results

### 3.1. Descriptive Statistics

Descriptive statistics are presented in Table 1. The mean level of the frequency of social media use at T1 was 23.01 (SD = 8.10), and on average, adolescents reported one symptom of problematic social media use at T1. At T1, more than half of the participants (65.8%) reported going to sleep at 10.00 p.m. or later. At T2, this percentage increased to 81.8%. The average score of perceived quality of sleep was 3.32 (SD = 0.92) at T1 and 3.35 (SD = 0.89) at T2. At T1, about half of the adolescents indicated being rarely or never allowed to use the Internet or play games in the hour before going to sleep (53.3%) or take the smartphone/tablet to the bedroom when going to sleep (46.2%).

### 3.2. Correlations

Correlations between all variables of interest can be found in Appendix A. Both the frequency of social media use and problematic social media use were correlated significantly and positively with bedtime and negatively with perceived quality of sleep. This indicates that more frequent use of social media and more problematic social media use indicated later bedtime and a poorer quality of sleep. In addition, strict parental rules about using the Internet in the hour before going to sleep and taking the smartphone to the bedroom when going to sleep were significantly related to less frequent social media use, fewer symptoms of problematic use, earlier bedtime, and better quality of sleep. Finally, boys reported higher sleep quality and a later bedtime compared to girls, and older adolescents reported a lower sleep quality (at T2 only) and a later bedtime compared to younger adolescents.

### 3.3. Social Media Use, Problematic Social Media Use, and Sleep

The model testing the effects of the frequency of social media use on sleep outcomes showed an acceptable fit to the data (CFI = 0.998, RMSEA = 0.03). The frequency of social media use at T1 had a significant effect on bedtime at T2 (Table 2), indicating that more frequent use of social media predicts later bedtime one year later (*β* = 0.14, *p* < 0.001). The frequency of social media use did not significantly predict the perceived quality of sleep.

The model testing the effects of problematic social media use also showed an acceptable fit to the data (CFI = 0.997, RMSEA = 0.03). The effect of problematic social media use at T1 on bedtime at T2 (Table 2) was significant, implying that more problematic social media use predicted later bedtime one year later (*β* = 0.07, *p* < 0.01). The effect of problematic social media use on the perceived quality of sleep was non-significant.

When we tested the effects of the frequency of social media use and problematic social media use simultaneously (the combined model; Table 2), the model fit was also acceptable (CFI = 0.998, RMSEA = 0.03), but only the effect of the frequency of social media use on bedtime one year later was significant (*β* = 0.13, *p* < 0.01). The previously found effect of problematic use on later bedtimes was not significant (*β* = 0.04, *n.s.*).

### 3.4. The Moderating Role of Parental Rules

Four out of eight interaction effects between social media use and parental rules about access to the Internet/smartphone before bedtime on adolescents’ sleep were (at least marginally) significant. First, parental rules about using the Internet in the hour before going to sleep at T1 moderated the effect of the frequency of social media use at T1 (*β* = −0.17, *p* < 0.01) as well as the effect of problematic social media use at T1 (*β* = −0.24, *p* = 0.10) on the perceived quality of sleep at T2 (Figure 1A,B). In addition, parental rules about using the Internet in the hour before going to sleep at T1 moderated the effect of the frequency of social media use on bedtime at T2 (*β* = 0.12, *p* < 0.05; Figure 2A). Finally, parental rules about taking the smartphone to the bedroom when going to sleep at T1 moderated the effect of problematic social media use at T1 on the time adolescents go to sleep at T2 (*β* = 0.30, *p* < 0.05; Figure 2B). These findings indicated that strict parental rules seem to prevent subsequent sleep problems, i.e., low quality of sleep and delayed bedtime, particularly among adolescents who reported low levels of (problematic) social media use. Among adolescents who reported high levels of (problematic) social media use, strict parenting did not affect sleep quality or bedtime.

In the combined model that tested the effects of the frequency of social media use and problematic social media use simultaneously, three of the four previously found interactions were still (marginally) significant. We still found the moderating effect of rules about using the Internet in the hour before going to sleep at T1 on the relationship between frequency of social media use at T1 and quality of sleep (*β* = −0.13, *p* = 0.06) as well as on the relationship between problematic social media use at T1 and quality of sleep (*β* = −0.28, *p* = 0.06). Moreover, parental rules about taking the smartphone to the bedroom when going to sleep at T1 moderated the effect of problematic social media use at T1 on bedtime at T2 (*β* = 0.32, *p* < 0.05). The moderating effect of rules regarding Internet use in the hour before going to sleep on the relationship between social media use frequency and bedtime was no longer significant.

## 4. Discussion

The present study aimed to provide insight into the longitudinal relationship between social media use and adolescents’ sleep and clarify the parental role in protecting them against possible negative media influences. The longitudinal findings imply that more frequent use of social media and more problematic social media use predicted a later bedtime among adolescents one year later. However, the effects of frequency of social media use and problematic social media use on bedtime depended on the rules that parents set, i.e., strict parental rules on the Internet and smartphone use before going to sleep predicted an earlier bedtime, but only among adolescents who scored below average on the frequency of social media use and/or problematic social media use. In a similar vein, strict parental rules regarding Internet use in the hour before sleep predicted a better quality of sleep, but only among adolescents who scored below average on the frequency of social media use and/or problematic social media use. Among adolescents who used social media more frequently or reported more problematic social media use, strict parental rules did not predict a better quality of sleep and, therefore, did not prevent negative media influences on sleep.

Consistent with a cross-sectional study by Pieters et al. [48], the present longitudinal study suggests that restricting access to the Internet and smartphones before bedtime may be an effective parenting tool to improve adolescents’ bedtime habits as well as their sleep quality. In addition, the present study suggests that parental rules are only effective for adolescents who are less engaged with social media. Once adolescents become highly engaged social media users, restrictive parental rules no longer seem to have a protective effect on adolescents’ bedtime and sleep quality. A possible explanation for this finding may be that highly engaged social media users, including users who report problematic social media use, experience more stress and fear of missing out (FOMO [56]) when they are not allowed to check their social media late in the evening, for instance, because they experience a stronger urge to keep themselves updated all the time. In this regard, Cheever, Rosen, Carrier, and Chaves [57] showed that heavy daily users of mobile devices experienced more anxiety due to the fear of missing out when their access to these devices was denied compared to low and moderate daily users. Experiencing FOMO could prevent adolescents from falling asleep and/or lead to more restless sleep [58]. A second explanation relates to perceived social norms and expectations, as adolescents may not stop their social media use as they may experience fear of social disapproval when violating norms on steady online availability and prompt responding [59]. A third explanation may be that restrictive parental rules may provoke more conflicts with parents when adolescents are highly engaged in social media use and/or display signs of social media addiction. Mesch [60] showed that adolescents who use the Internet more frequently have more conflicts with their parents about such use. These parent–child conflicts (late in the evening) may evoke emotional insecurity [61] and physical and cognitive arousal, which may interfere with adolescents’ healthy sleep patterns. Therefore, more frequent and more problematic social media use seems to be associated with later bedtime and lower sleep quality. Moreover, strict rules about access to the Internet in the evening can protect against these negative influences, but only when adolescents have not yet developed a strong attachment to social media.

Both frequency of social media use and problematic social media use predicted a later bedtime. The relationship between frequency of use and bedtime remained significant after controlling for problematic social media use, indicating that this relationship is robust and does not result from the conceptual overlap between social media use intensity and problematic use. On the contrary, the relationship between problematic social media use and bedtime was non-significant when we controlled for the frequency of social media use. Nevertheless, we believe that this later finding is meaningful, as frequent use of social media can be regarded as a central characteristic of problematic use of social media, whereas intensive use of social media can easily exist without any signs of problematic use. Therefore, not finding a significant effect of problematic use on sleep after controlling for the frequency of use seems logical.

Finally, parental rules regarding social media use in the hour before going to sleep generated more convincing results than parental rules regarding taking a smartphone or tablet to the bedroom; therefore, they may be more effective in preventing sleep problems. These two parenting rules overlap, as preventing social media use in the hour before going to sleep also precludes using a smartphone or tablet inside the bedroom. Therefore, this parental rule may be more effective in improving bedtime and sleep quality in adolescents.

### Strengths and Limitations

This study’s strengths include a longitudinal design with a one-year follow-up measurement and strict statistical tests in which we controlled for previous levels of the outcome variables and demographic factors. However, some limitations should be noted. First, data were based on adolescents’ self-reports. It would be valuable for future studies to use objective measures of social media use and sleep habits. Second, there are significant differences between participants who dropped out and who remained in this longitudinal study. Although these differences can be understood in light of the fact that more classes in the last year of lower-level secondary education dropped out, these differences still limit the unbiasedness of the sample and the generalizability of results. Third, some interaction effects were borderline significant. Yet, given the values of the standardized coefficients, the rigorous statistical tests, and the long-time-interval, we believe that these interaction effects should be considered meaningful, warranting further study. Finally, we did not assess factors such as (mental) health problems and caffeine and nicotine consumption, which are known to affect adolescents’ sleep [62,63,64]. Therefore, future research should also include these factors.

## 5. Conclusions

As far as we know, this is the first study to test the longitudinal effects of (problematic) social media use on adolescents’ sleep and to model the potential buffering role of parental rules about Internet/smartphone use before going to sleep. The findings suggest that the intensity of social media use and the degree of problematic social media use play a critical role in adolescents’ tendency to delay their bedtime. Restricting access to the Internet and smartphones before going to sleep seems to be an effective way to optimize adolescents’ sleep time. This, however, is only the case if adolescents are not too engaged with social media. Once adolescents have developed these risky social media use patterns, strict parental rules regarding Internet and smartphone use before going to sleep no longer seem to influence bedtime and sleep quality. The current study extends our understanding of differential susceptibility to media effects [65] by indicating that the effect of social media use on sleep differs from adolescent to adolescent, and that these effects are not only dependent upon individual characteristics, but also on contextual factors such as parenting behaviors. More practically, based on these findings, parents should be advised to prohibit their children from using the Internet and smartphones in the evenings before they become highly involved in social media use or display symptoms of problematic social media use. Whether these parental rules will help prevent children from becoming highly involved social media users is not completely clear yet [46,47] and is an important question that should be addressed in future research.

## Figures and Tables

**Figure 1 ijerph-18-01346-f001:**
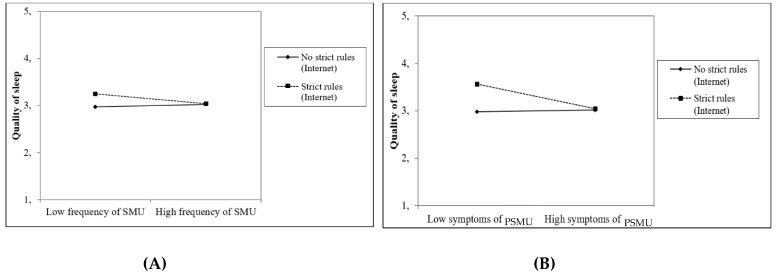
Moderation of the effect of the frequency of social media use (SMU; (**A**)) and problematic social media use (PSMU; (**B**)) on perceived quality of sleep by parental rules about using the Internet in the hour before going to sleep.

**Figure 2 ijerph-18-01346-f002:**
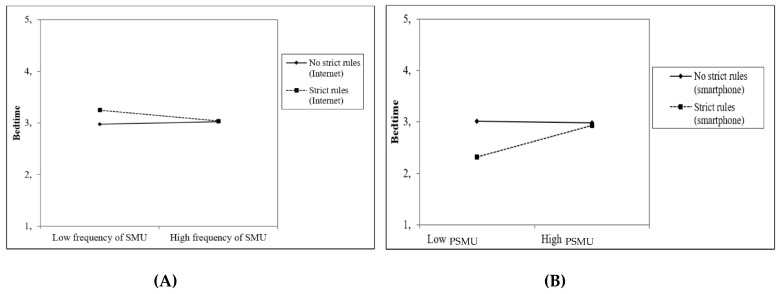
Moderation of the effect of frequency of social media use (SMU) and problematic social media use (PSMU) on bedtime by parental rules about using the Internet in the hour before going to sleep (**A**) and about taking the smartphone to the bedroom when going to sleep (**B**).

**Table 1 ijerph-18-01346-t001:** Descriptive statistics of study variables.

Total *N* = 2021	*n* (%)	M (SD)	Min	Max
Gender (girls)	917 (45.4%)			
Educational level (higher)	1169 (57.8%)			
Age T1		13.86 (1.25)	11	17
Frequency social media use T1		23.01 (8.10)	4	42
Problematic social media T1		10.25 (1.54)	9	18
Bedtime T1		6.49 (2.09)	1	10
Bedtime T2		7.17 (1.89)	1	10
Quality of sleep T1		3.32 (0.92)	1	5
Quality of sleep T2		3.35 (0.89)	1	5
Parental rules Internet T1 (strict)	1078 (53.3%)			
Parental rules smartphone T1 (strict)	934 (46.2%)			
Non-responders at T2	608 (30%)			

**Table 2 ijerph-18-01346-t002:** Structural equation model of the effects of the frequency of social media use and problematic social media use separately and jointly on bedtime and perceived quality of sleep.

	Bedtime T2		Quality of Sleep T2	
Frequency of Social Media Use	β	SE	β	SE
Gender (ref = girls)	0.09 **	0.02	0.04	0.02
Age	0.15 **	0.03	−0.08 *	0.03
Educational level (ref = higher)	−0.02	0.03	0.01	0.02
Bedtime T1	0.49 **	0.03	-	-
Quality of sleep T1	-	-	0.58 **	0.02
Frequency social media use T1	0.14 **	0.03	−0.04	0.02
Problematic Social Media Use	β	SE	β	SE
Gender (ref = girls)	0.07 **	0.02	0.04	0.02
Age	0.15 **	0.03	−0.08 *	0.03
Educational level (ref = higher)	−0.01	0.03	0.01	0.02
Bedtime T1	0.52 **	0.03	-	-
Quality of sleep T1	-	-	0.58 **	0.02
Problematic social media use T1	0.07 *	0.02	−0.03	0.02
Frequency of Social Media Use and Problematic Social Media Use	β	SE	β	SE
Gender (ref = girls)	0.09 **	0.02	0.04	0.02
Age	0.15 **	0.03	−0.08 *	0.03
Educational level (ref = higher)	−0.02	0.03	0.01	0.02
Bedtime T1	0.49 **	0.03	-	-
Quality of sleep T1	-	-	0.58 **	0.02
Frequency social media use T1	0.13 **	0.03	−0.03	0.02
Problematic social media use T1	0.04	0.03	−0.03	0.03

Note. *β* = standardized coefficient; SE = standard error. * *p* < 0.01. ** *p* < 0.001.

## Data Availability

The data presented in this study are available on request from the corresponding author. The data are not publicly available yet because, at the time if proofreading, we were stil in the process of making the data publicly available in an accessible repository.

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
