# Peer review of "Social Media Use and Adolescents’ Sleep: A Longitudinal Study on the Protective Role of Parental Rules Regarding Internet Use before Sleep"

_ijerph, 2021, doi:10.3390/ijerph18031346_

Round 1
Reviewer 1 Report
- Suggest adding meaning to the statistic numbers (M and SD) in the abstract.
- Page 3 of 15, last paragraph, “These relationships will be tested …”, suggest changing to “These relationships are tested …”.
- Page 4 of 15, the first paragraph, “as some variables of interest were not included in the questionnaire in previous waves,” this sentence is confusing. Which previous waves are referred to?
- Page 4 of 15, the second paragraph, “Of the 2.708 participants included at T1 …”, should it be “of the 2,708 …”? The same for the “a sample of 2.021 participants eligible for …”, “a sample of 2,021 …”? A comma instead of a decimal.
- Page 4 of 15, the third paragraph, rather than just listing the statistic numbers, suggest adding some explanations to the numbers.
Reviewer 2 Report
Dear authors,
Thank you for the opportunity of reviewing this interesting work, following I give some minor suggestions/questions:
- Did you have missing data? How did you manage them?
- Did you explore the cluster bias? How did affect to the results?
- Why did you use SEM instead of regression analyses?
- Please, include the missing data in tha table 1.
- There are many tables/figures. I suggest to move the table 2 to supplementary material. Table 3 to 5 must be combined with focus in the main predictors. Moreover, the order of the tables in relation to the text is confusing.
- The moderation was tested in different models than those that appear in the tables? This is not clear for me. If yes, why did you perform separated models? If not, why do you not included the interation effect in the tables? Finally, all the interation effects can be shown in an only figure, and, please, include significance test (for example, simple comparisons) in the figure.
- Part of the information that is given in the discussion, specifically the first paragraph line 6-9 must be moved to the results section.
Reviewer 3 Report
This study explored that effects of the frequency of social media use and symptoms of social media addiction on adolescents’ bedtime and quality of sleep, and examined the moderating effects of parental rules regarding Internet and smartphone use one hour before sleep. The topic is interesting, results are valuable from both theoretical and practical perspectives. However, there are some questions needing to be addressed. My detailed suggestions are as follows:
- The theoretical basis is weak in this study, specially, why the parental rules regarding Internet and smartphone use one hour before sleep can moderate the effects of the frequency of social media use and symptoms of social media addiction on adolescents’ bedtime and quality of sleep, and what is the relationship between parental rules and mobile phone use, and why authors combined them together as moderating variables. There need more words to cover those issues.
- Previous studies have demonstrated that frequency of social media use could predicted symptoms of social media addiction (Amichai-Hamburger & Furnham, 2007; Caplan, 2003; Fioravanti et al., 2012; McKenna & Bargh, 2000; Smahel et al., 2012; Yu et al., 2017) However, in this study, authors considered them as two independent variables, which make me confusing. I want authors to make a clear statement on this relationship.
Amichai-Hamburger, Y., & Furnham, A. (2007). The positive net. Computers in Human Behavior, 23(2), 1033-1045. doi: 10.1016/j.chb.2005.08.008
Caplan, S. E. (2003). Preference for online social interaction: A theory of problematic Internet use and psychosocial well-being. Communication Research, 30(6), 625-648. doi: 10.1177/0093650203257842
Fioravanti, G., Dettore, D., & Casale, S. (2012). Adolescent Internet addiction: Testing the association between self-esteem, the perception of Internet attributes, and preference for online social interactions. Cyberpsychology Behavior and Social Networking, 15(6), 318-323.
Smahel, D., Brown, B. B., & Blinka, L. (2012). Associations between online Friendship and Internet addiction among adolescents and emerging adults. Developmental Psychology, 48(2), 381-388. doi: 10.1037/a0027025
Yu Q., Zhang L., Wu S., Guo Y., Jin S., & Sun Y. (2017). The Influence of Juvenile Preference for Online Social Interaction on Problematic Internet Use: The Moderating Effect of Sibling Condition and the Moderated Moderating Effect of Age Cohort. Computers in human behavior, 68, 345-351.doi: 10.1016/j.chb.2016.11.026
- The sample was presented confusing, please check it.
- There are wild and significant difference between the dropped participants and remained participants, which would limit the unbiasedness of sample and generalizability of results. This issue should be discussed in the section of discussion.
- In the section of result, some statements need further modify, e.g., “Among adolescents reporting relatively high social media use or social media addiction symptoms, parental strictness did not affect sleep quality or bedtime.” parental strictness was considered as moderating variable, rather than independent variable, thus, this statement should be modified into “Among stricter parental rules, social media use or social media addiction effected ……”
- I suggest authors present the results of the moderating effect analysis in a table, which would be more conducive to readers' reading
- The discussion, especially the theoretical significance, needs to be strengthened.
Author Response
Reviewer#3
- The theoretical basis is weak in this study, specially, why the parental rules regarding Internet and smartphone use one hour before sleep can moderate the effects of the frequency of social media use and symptoms of social media addiction on adolescents’ bedtime and quality of sleep, and what is the relationship between parental rules and mobile phone use, and why authors combined them together as moderating variables. There need more words to cover those issues.
The theoretical rationale for the moderating effect of parental rules regarding internet and smartphone use is given in the first paragraph of section 1.3.: “It has been argued that the timing rather than the duration of social media use may affect adolescents’ sleep [15], as social media use at bedtime is a strong predictor of poorer sleep outcomes [37]. Thus, social media use during the evening hours and before going to sleep may particularly be associated with sleep problems. Therefore, it can be expected that the rules that parents set regarding Internet and smartphone use before sleep will buffer the assumed negative effect of social media use on sleep. For instance, it can be anticipated that adolescents who are allowed to bring their smartphone or tablet to their bedroom when they go to sleep will experience more sleep problems than adolescents who have to leave their smartphone or tablet behind.”
We did not combine two variables. The focus of this study is on the potential protective role of parental rules regarding social media use before bedtime. Because we assumed that social media use during the hour before bedtime particularly takes place through smartphone or tablet, two devices that can very easily be used in bed, one of the two items measuring parental rules explicitly asked whether parents allow participants to take their smartphone or tablet to their bedroom when they go to sleep. We hope that this point is clearer now.
- Previous studies have demonstrated that frequency of social media use could predicted symptoms of social media addiction (Amichai-Hamburger & Furnham, 2007; Caplan, 2003; Fioravanti et al., 2012; McKenna & Bargh, 2000; Smahel et al., 2012; Yu et al., 2017) However, in this study, authors considered them as two independent variables, which make me confusing. I want authors to make a clear statement on this relationship.
We agree that frequency of social media use and symptoms of social media addiction are related concepts. In general, the found correlations between these two concepts, however, are weak to moderate (Boer et al, 2019; Van den Eijnden et al. 2018). An increasing number of researchers stress that this effect size is plausible, because the frequency of social media use and symptoms of social media addiction should be regarded as related but different dimensions (Boer et al, 2019; Scott & Woods,2019; Van den Eijnden et al. 2018;).
To make this point more clearly, we have edited the text: “Moreover, several scholars have emphasized that frequency of social media use and problematic social media use should be regarded as related but different dimensions [13,14,15]. For instance, frequent users may be well able to regulate their social media use, whereas users who developed symptoms of social media addiction may not show intensive social media use due to a mismatch between their desired and actual online social network, which may cause preoccupation with social media without having the desired network to interact with. Therefore, we differentiated between the frequency of social media use and problematic social media use, further referred to as social media addiction symptoms.”
Scott, H.; Woods, H. C. Understanding links between social media use, sleep and mental health: Recent progress and current challenges. Current Sleep Medicine Reports, 2019, 5, 141–149. https://doi.org/10.1007/s40675-019-00148-9.
Boer, M.; Stevens, G.; Finkenauer, C.; Van den Eijnden, R. ADHD-symptoms, social media use intensity, and social media use problems in adolescents: Investigating directionality. Child Development, 2019 91(4), pp.e853-e865. https://doi.org/10.1111/cdev.13334.
Van Den Eijnden, R. J. J. M.; Koning, I. M.; Doornwaard, S.; Van Gurp, F.; & Ter Bogt, T. The impact of heavy and disordered use of games and social media on adolescents’ psychological, social, and school functioning. Journal of Behavioral Addictions, 2018, 7(3), 697-706. https://doi.org/10.1556/2006.7.2018.65
- The sample was presented confusing, please check it.
We checked the description of the sample and edited it slightly. Hopefully, the edited version of the sample description is clearer now.
- There are wild and significant difference between the dropped participants and remained participants, which would limit the unbiasedness of sample and generalizability of results. This issue should be discussed in the section of discussion.
In the method section we have added the following information in the section describing differences between participants who dropped out and who remained: “These differences may to some extent be explained by the fact that several whole school classes from the last year of lower-level secondary education dropped out.”
In the discussion section, the following limitation was added: “Second, there are significant differences between participants who dropped-out and who remained in this longitudinal study. Although these differences can be understood in the light of the fact that more classes in the last year of lower-level secondary education dropped out, these differences still limit the unbiasedness of the sample and the generalizability of results.”
- In the section of result, some statements need further modify, e.g., “Among adolescents reporting relatively high social media use or social media addiction symptoms, parental strictness did not affect sleep quality or bedtime.” parental strictness was considered as moderating variable, rather than independent variable, thus, this statement should be modified into “Among stricter parental rules, social media use or social media addiction effected ……”
We see this point by reviewer#3 and this statement has been changed into: “Among adolescents with stricter parents, social media use and symptoms of social media addiction did not affect sleep quality or bedtime.”
- I suggest authors present the results of the moderating effect analysis in a table, which would be more conducive to readers' reading
This would mean that Table 3 would have 12 extra lines. For the clarity of data presentation, we, therefore, decided to describe the relevant statistics for these 12 interaction terms in the text and add 4 figures to provide insight into the meaning of these interactions. We tried to depict all 4 interactions in one Figure, as was suggested by reviewer#2. However, this figure was clogged and very difficult to read. We, therefore, decided to stick to the 4 original figures and present them in two integrated figures (Figure 1 and Figure 2).
- The discussion, especially the theoretical significance, needs to be strengthened.
We have added the following theoretical notion to the conclusion section of the discussion to strengthen the theoretical significance of the findings: “The current study extends our understanding of differential susceptibility to media effects (63) by indicating that the effect of social media use on sleep differ from adolescent to adolescent, and that these effects are not only dependent upon individual characteristics, but also on contextual factors such as parenting behaviors.”
Valkenburg PM, Peter J. The differential susceptibility to media effects model. Journal of communication. 2013 Apr 1;63(2):221-43.